# Welfare Indices in Anurans under Human Care

Ethel Cortés Pérez [1] and Ricardo Itzcóatl Maldonado Reséndiz [2,*]

1    Zoological Medicine Preparatory Program, Facultad de Medicina Veterinaria y Zootecnia, Universidad Nacional Autónoma de México, Mexico City 04510, Mexico; copeeth@gmail.com
2    Ethology, Wildlife and Laboratory Animal Department, Facultad de Medicina Veterinaria y Zootecnia, Universidad Nacional Autónoma de México, Mexico City 04510, Mexico
*    Correspondence: itzcoatl.maldonado@fmvz.unam.mx

**Abstract:** Certain species within the order Anura are relatively new in the context of exotic animals as pets, and the precise conditions required for their optimal care and well-being are still not well understood. This knowledge gap highlights the crucial need to develop effective strategies to measjournal oure the welfare of these animals. To address this need, the objective of this study was to review the existing literature on welfare indices related to amphibians kept under human care. A systematic review was conducted across eight scientific databases, with a focus on identifying relevant articles that explored welfare indices specifically within the order Anura. The search was performed using two specific keywords. In total, 1568 English language results were obtained. Following a refinement process, 19 articles were selected for further analysis. The most investigated welfare indices in amphibians included individual survival, life expectancy, reproduction, hibernation, and body condition. However, it is important to note that studies specifically examining the behavior of amphibians within the order Anura are limited in scope. It is evident that there is still much work to be conducted in order to gain a deeper understanding of the environmental conditions and cognitive processes that affect the well-being of these animals.

**Keywords:** Anuran; welfare indices; well-being

## 1. Introduction

Animal welfare encompasses the well-being of individual animals and their ability to effectively adapt to their environment. While some instances of welfare can be achieved with minimal effort and resources, reflecting satisfactory conditions experienced by the animals, there are situations where animals encounter difficulties in coping with their environment, resulting in compromised welfare [1]. Moreover, the concept of "animal welfare" encompasses three essential aspects: physical health, emotional state, and behavior. The first aspect entails the absence of disease, injuries, and ensuring adequate nutrition, physical comfort, and thermal conditions. The second aspect emphasizes the absence of negative emotions, such as pain, stress, and fear, contributing to a positive emotional state. Lastly, behavior serves as a significant parameter to assess animal welfare. Observations and interpretations of behavior are crucial as they can provide insights into emotional states, which can be either negative or positive [2,3].

Mexico is recognized as the fourth-ranking country in terms of amphibian diversity, boasting a total of 361 species that are classified into three distinct orders: Anura, Caudata, and Gymnophiona. These amphibians display a wide distribution across diverse biomes, showcasing remarkable adaptability despite their reliance on water [4,5]. Notably, they possess a distinctive feature whereby oxygen is extracted principally through their lungs during the adult stage and through gills in their aquatic larval stage. Furthermore, their skin, permeable to water, plays a vital role in respiration while also serving as a protective barrier against pathogens. Amphibians, unlike other animal groups, lack inherent physiological mechanisms when regulating body temperature, which, in turn, has shaped their geographic distribution, predominantly favoring temperate and warm climate regions [4].

In the southeastern region of Mexico, amphibians are sporadically utilized primarily for medicinal purposes [5]. In communities where frog species inhabit natural bodies of water or water systems, it is common to find them in residential areas, including gardens, fountains, pools, and even cisterns. Some people perceive these encounters as an opportunity to keep them as pets [6]. In the central part of the country, Environmental Management Units (UMAS) are prevalent and engage in the utilization of amphibians for various purposes. It is important to note that these activities predominantly involve exotic species and do not directly involve native fauna [5].

The Secretaría de Pueblos y Barrios Originarios y Comunidades Indígenas Residentes de la Ciudad de México actively promotes projects focused on the production of bullfrogs (*Lithobates catesbeianus*) for the restaurant industry. Bullfrogs have been a traditional food source since pre-Hispanic times and are renowned for their rich content of omega-3 and omega-4 fatty acids. Notably, they are also free of cholesterol and can attain weights of up to half a kilogram [7].

Amphibians produced in captivity also serve as popular companion animals. Commonly traded species for this purpose include axolotls (*Ambystoma* sp.) and certain tree frogs, such as red-eyed tree frogs (*Agalychnis callidryas*) or the green tree frog (*Pachymedusa dacnicolor*) [5]. The extent of amphibian ownership in Mexico has not been adequately evaluated. However, data from the AVMA PET OWNERSHIP AND DEMOGRAPHICS SOURCEBOOK reveals a reported 25.5% increase in exotic pet ownership, including amphibians, in the United States from 2011 to 2016 [8].

Considering the above, it is worth noting that certain species of amphibians, particularly Anurans, are emerging as novel companion animals, yet the exact requirements for their optimal care remain uncertain [1,9]. The objective of this systematic review was to identify welfare measurement strategies that could assess the well-being status of Anurans when kept in captivity.

## 2. Materials and Methods

A comprehensive systematic review was conducted, focusing on the welfare indices and pathological aspects associated with stressful conditions in amphibians of the order Anura. The search encompassed the following reputable electronic databases: Google Scholar, BIDI UNAM, Redalyc, Scopus, PubMed, Mendeley, Scielo, and Web of Science. The search terms employed were "welfare of captive amphibians," "enrichment amphibians," "bienestar en anfibios" (well-being in amphibians), and "enriquecimiento en anfibios" (enrichment in amphibians).

The following refinement criteria were applied to the search results: articles that involve species of the order Anura were included, obtaining 1315 as a result and excluding 253. Articles that involved issues relating to behavior were included, with 73 documents remaining and 1242 excluded. Articles whose texts were found to be complete included 27 articles, while 8 were excluded.

## 3. Results

A total of one thousand five hundred and sixty-eight English language results were found, with 114 for the keyword "welfare of captive amphibians" and 1454 for "enrichment amphibians." Subsequently, only those discussing the order Anura were selected ($n$ = 1315), followed by those addressing behavior ($n$ = 73). Additionally, only articles available in their full text were included ($n$ = 27), and duplicate results were excluded (Figure 1), resulting in 19 condensed articles (Table 1).

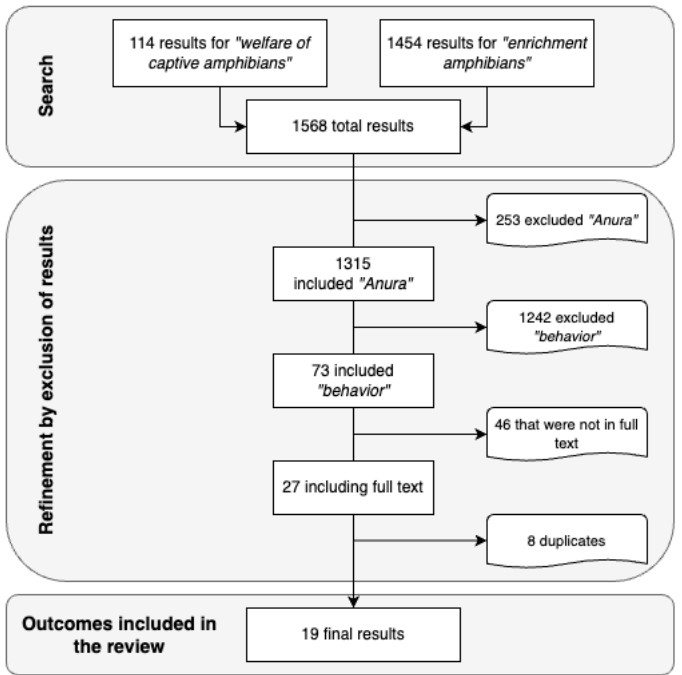

**Figure 1.** Flowchart of the systematized search.

**Table 1.** General information about the articles included in the systematic review.

| References | Taxon | Topic and Main Idea |
|---|---|---|
| Hayes M. et al. Beyond mammals. Second Nature: Environmental Enrichment for Captive Animals. 1998 [10] | N/A | Animal welfare: identify enrichment programs through three requirements:<br>1. Contact with congeners;<br>2. Interaction with other species;<br>3. Physical characteristics of the captive environment. |
| Hurme K. et al. Environmental enrichment for dendrobatid frogs. Appl. Anim. Welf. Sci. 2003 [11] | *Dendrobates* sp. | Environmental enrichment: identify animal welfare strategies for the development of dendrobate hatchlings |
| Archard G. Effect of enrichment on the behaviour and growth of juvenile *Xenopus laevis.* Appl. Anim. Behav. Sci. 2012 [12] | *Xenopus laevis* | Environmental enrichment: determine if environmental enrichment influences the development of *Xenopus laevis* |
| Burghardt G. Environmental enrichment and cognitive complexity in reptiles and amphibians: Concepts, review, and implications for captive populations. Appl. Anim. Behav. Sci. 2012 [13] | N/A | Environmental enrichment: importance of well-being in reptile and amphibian collections |
| Chum H. et al. Biology, behavior, and environmental enrichment for the captive African clawed frog (*Xenopus* spp). Appl. Anim. Behav. Sci. 2013 [14] | *Xenopus laevis* | Environmental enrichment: strategies for environmental enrichment |
| Harrrington L. et al. Conflicting and complementary ethics of animal welfare considerations in reintroductions. Conserv. Biol. 2013 [15] | N/A | Conservation: animal welfare strategies for species when released |

**Table 1.** *Cont.*

| References | Taxon | Topic and Main Idea |
| --- | --- | --- |
| Cikanek S. et al. Evaluating group housing strategies for the ex-situ conservation of harlequin frogs (*Atelopus* spp.) using behavioral and physiological indicators. Plos One. 2014 [16] | *Atelopus certus-Atelopus glyphus* | Conservation: stress evaluation in a group of frogs through aggressive behaviors and corticosteroids |
| Michaels C. et al. Impact of plant cover on fitness and behavioral traits of captive red-eyed tree frogs. (*Agalychnis callidryas*). Plos one. 2014 [17] | *Agalychnis callidryas* | Conservation: implementation of plants in frog enclosures for their conservation |
| Michaels C. et al. The importance of enrichment for advancing amphibian welfare and conservation goals: A review of a neglected topic. Amphib. Reptile Conserv. 2014 [18] | N/A | Environmental enrichment: enrichment for the stimulation of captive amphibians |
| Beltrán N. et al. Guía de anfibios de los parques nacionales españoles. Red de parques nacionales. 2016 [4] | N/A | Conservation: viology of the species |
| Manteca X. et al. Animal-based indicators to assess welfare in zoo animals. CAB Rev. 2016 [19] | N/A | Animal welfare: identify the welfare indicators in zoological collections of reptiles and amphibians |
| Grant R. et al. ExNOTic: Should we be keeping exotic pets?. Anim. 2017 [20] | N/A | Animal welfare: to question whether we should keep exotic animals as pets |
| Pasmans F. et al. Future of keeping pet reptiles and amphibians: towards integrating animal welfare, human health and environmental sustainability. Vet. Rec. 2017 [21] | N/A | Animal welfare: responsible ownership of amphibians and reptiles as pets |
| Brod S. et al. Discussing the future of amphibians in research. Lab. Anim. 2019 [13] | Amphibians | Animal welfare: identify the guidelines for keeping amphibians under human care |
| Ferrel S. et al. Amphibian behavior for the exotic pet practitioner. Vet. Clin. Exot. Anim. 2020 [22] | N/A | Behavior: examine the highlights of amphibian well-being, pain, and behavior |
| Testud G. et al. Acoustic enrichment in wildlife passages under railways improves their use by amphibians. Glob. Ecol. Conserv. 2020 [23] | *Triturus* spp. | Conservation: evaluation of the impact of transport infrastructure on free-living specimens |
| Woody S. et al. Posture as a non-invasive indicator of arousal in American toads (*Anaxyrus americanus*). J. Zool. Bot. Gard. 2020 [24] | *Anaxyrus americanus* | Animal welfare: the relationship between the posture of the specimens with the excitement caused by stress in individuals |
| Ramos J. et al. Evaluation of environmental enrichment for *Xenopus laevis* using a preference test. Lab. Anim. 2021 [25] | *Xenopus laevis* | Environmental enrichment: examine frog preferences for different items of enrichment |
| Hollew T. et al. Pet management practices of frog and turtle owners in Victoria, Australia. Vet. Record. 2022 [26] | N/A | Animal welfare: to investigate if the owners of frogs and turtles are aware of their welfare needs |

## 4. Discussion

The results obtained from the systematic review show three main elements within the content: (1) animal welfare, (2) conservation, and (3) environmental enrichment. Within these, topics such as the biology of the species, the implementation of enriched environments, and responsible ownership of amphibian specimens are addressed.

### 4.1. Knowledge about Behavior in Anurans

For a long time, amphibians and other ectothermic species have been perceived as animals that "cannot suffer" or "do not feel pain," at least not to the same extent as mammals and birds. This perception has led to a bias in the development of animal welfare plans for species in this group [18]. As a result, the management of captive amphibians primarily focuses on meeting their basic survival requirements, such as hydration, temperature, and food [13,18,20].

However, notable behaviors have been documented within the order of Anura, including communication abilities, habituation, avoidance, and spatial orientation. For instance, some toads can effectively avoid toxic plant solutions once identified, and they also exhibit the ability to memorize mazes [13]. Tungara frogs (*Engystomops pustulosus*) demonstrate a capacity to adjust their vocalizations when communicating with conspecifics or predators. In species like *Theloderma corticale* (Hylidae), repetitive behaviors, such as letting themselves be carried by air currents, have been observed, possibly associated with play behavior. Additionally, harmless fights have been observed in poison dart frogs (Dendrobatidae species) [13].

It is important to acknowledge that there are no "easy to maintain" amphibians [20], and several studies have shown that many companion animals do not always experience a good level of welfare. Research on the behavioral and physiological indicators reflecting this state in the order Anura has revealed that they may not be effective [27]. Furthermore, the rapid deterioration of many amphibians due to poor environmental conditions prevents the presentation of behavioral manifestations that could allow us to identify any level of suffering prior to death. Another limitation is the reduced activity and lower metabolic capacity of these animals, which can mask or reduce the occurrence of repetitive (stereotyped) behaviors in certain taxa [18].

### 4.2. Welfare Indexes

It is essential to recognize that assessing animal welfare should consider both the physical and mental health of animals, as well as their behavior [18,19,22]. Welfare indicators can be categorized as either environment-based or animal-based. Environment-based indicators often offer the advantage of easier measurement. However, specific environments can exhibit parameter variability, making their impact on individual welfare unpredictable. Additionally, this effect depends on the unique characteristics and interactions of each animal with the environment [19].

Despite this, welfare indices used in amphibians have traditionally focused on the physiology and health status of individuals, often resulting in specimens being unable to cope with their environment or experiencing developmental consequences due to exerted efforts [1]. Harrington et al. [15] documented the welfare indicators most frequently used in captive settings. They found that common indicators for assessing amphibian welfare included individual survival, life expectancy, reproduction, hibernation, and body condition [15].

Torpor is defined as abnormal inactivity and a lack of interest in environmental stimuli. The characteristics of this behavior include reduced interaction with conspecifics, limited mobility, and decreased exploration of novel objects. Torpor can occur when individuals have a perceived sense of lack of control over their environment and is considered a low welfare indicator [19].

Survival is associated with the prevalence and incidence of diseases; the presence of pathologies implies pain, itching, or other forms of discomfort in animals, which results in a decrease in welfare due to the stress generated [19].

When the lifespan falls below expectations, it may indicate that the animal has experienced stress and, consequently, has poor welfare. However, it is important to note that this indicator has limitations when assessing the current welfare status of individuals or groups. It should be acknowledged that a long life does not necessarily guarantee a good quality of life in terms of welfare. For instance, an older animal may have a compromised quality

of life, whereas the death of a young animal may not be linked to inadequate welfare conditions [19].

Significant weight loss and poor body conditions often indicate the presence of disease or an individual's inability to meet their nutritional requirements. This can be due to an inadequate diet or the animal's inability to compete with others for access to food [19].

The ability of individuals to reproduce has been considered an indicator of welfare since good health is one of the requirements for reproductive success. In most cases, an animal that is unable to reproduce may have underlying conditions associated with disease [12,22].

Prolonged or intense stress directly affects the welfare of individuals. Stress involves the activation of the hypothalamic–pituitary–adrenal axis, leading to an increase in glucocorticoid secretion. Therefore, the concentration of glucocorticoids has also been used to assess welfare in amphibians, and this can be measured in plasma, saliva, feces, and urine [16,19].

Furthermore, behavior has more recently been considered an indicator of welfare, as in the case of *Xenopus laevis* frogs, where their activity in water and their preferred locations within a particular habitat have been measured [27].

Ethology is one of the most valuable tools for obtaining information about the behavior of any species, whether in captivity or in the wild [13]. The foundation of this science lies in ethograms, which are a list of specific behaviors that describe the elements and functions of each behavior through systematically scheduled observations with defined frequency and duration. Ethograms aid in providing data to identify the potential causes of problematic behavior [14]. In anurans, there is information on this. Cikanek et al. [16] published an article in which they reported the extraction of 66 frogs belonging to the species *Atelopus certus* and *A. glyphus* from their habitat to evaluate aggressive behaviors among them and measure glucocorticoid levels in their feces over a period of 5 weeks. The aggressive behaviors were categorized as follows: (a) fighting, (b) amplexus (mating behavior), (c) release call, (d) direct contact, (e) stalking, and (f) greeting (Figure 2).

At the end of the study, it was concluded that aggressive behaviors and the glucocorticoid levels detected in feces decreased over the weeks, suggesting that the frogs were able to adapt to their conspecifics. Therefore, fecal glucocorticoids could serve as a non-invasive measure for assessing stress in amphibians [16].

By contrast, Woody et al. [24] conducted a study to characterize the body postures of a group of toads (*Anaxyrus americanus*) and examined their association with throat expansion after visual stimulation using videos. The assumption was that higher postures and greater throat expansions could indicate heightened excitement in toads. Surprisingly, the results showed no significant correlation between the observed postures and throat expansion, suggesting that respiration might play a role in the process [24].

The information generated so far is limited to the Anura order; however, future research could evaluate whether there is a correlation between the presentation of certain behaviors and the well-being of individuals.

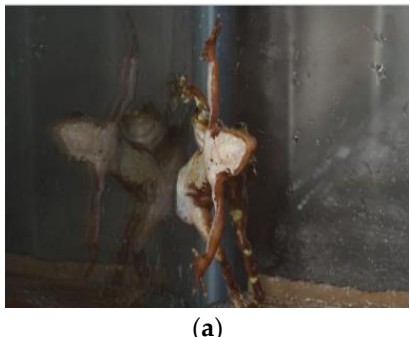    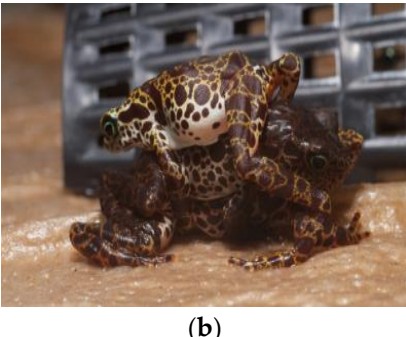

(**a**)                                                        (**b**)

**Figure 2.** *Cont.*

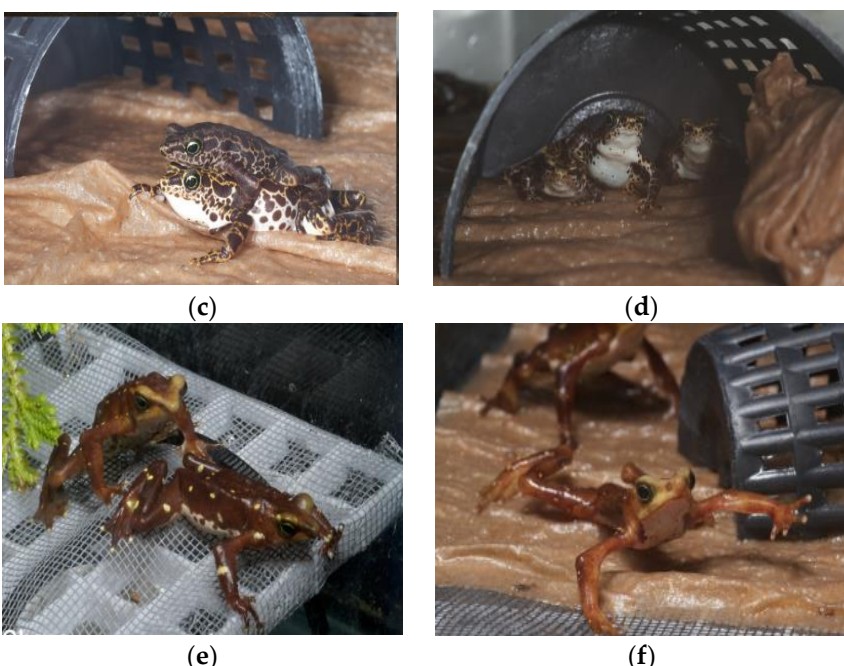

**Figure 2.** Aggressive behaviors described by Cikanek et al. [14] (9): Fight (**a**), Mount (**b**), Release call (**c**), Direct contact (**d**), Stalking (**e**) and Greeting (**f**).

*4.3. Environmental Enrichment*

Environmental enrichment includes any process that manipulates the environment of captive animals to enhance their welfare [17]. Ways in which an animal's environment can be enriched include the following:

(1) Managing a dynamic habitat;
(2) Encouraging social interactions between individuals;
(3) Encouraging foraging behavior;
(4) Introducing new objects;
(5) Species conditioning [11].

Although environmental enrichment methods have gained popularity, enrichment programs specifically designed for amphibians are rare [10]. Amphibian behaviors are closely tied to specific physiological functions like reproduction, basking, hunting, or burrowing, for which there has been limited enrichment research [18].

Providing shelter has been the most commonly employed form of environmental enrichment for frog species [18]. For instance, certain materials have been identified as preferred shelters for *Xenopus* spp., but studies have indicated no significant changes in body condition or growth rates associated with shelter [12,27]. Conversely, it has been demonstrated that shelters are crucial for the development of *Mannophryne trinitatis* and *Agalychnis callidryas* [18].

Another type of enrichment involves manipulating the substrate and water level. While *Mannophryne trinitatis* showed a preference for shallow bodies of water, *Engystomops pustulosus* and *Leptodactylus fuscus* favored gravelly substrates for burrowing [18]. Furthermore, *Dendrobates tinctorius* exhibited increased activity when provided with a coconut fiber substrate [11].

For species that actively forage, enrichment designs that stimulate food-searching efforts have been shown to elevate activity levels while reducing the risk of impaction and obesity [18].

Moreover, the use of acoustic environmental enrichment has been documented. Testud et al. [23] conducted a study involving specimens of *Pelophylax lessonae*, *P. ridibundus*, and other amphibians, where songs were played in tunnels to motivate individuals to cross to the other

side, yielding successful results. This opens up the possibility of utilizing frog vocalizations for enrichment purposes in captivity [23].

It is important to note that enrichment has been demonstrated to reduce mortality and injuries in certain amphibians while also yielding positive effects on growth rates and body condition indices. However, beyond physical health and well-being aspects, enrichment can also have implications for psychological well-being, leading to enhanced cognitive engagement [18].

*4.4. Consequences of Poor Well-Being*

Like other animal clades, Anuran amphibians kept in captivity exhibit a range of problems, not only due to poor environmental conditions but also as a result of behavioral conditions [14,21,26].

One of the main behavioral issues can be related to the establishment of hierarchies by territorial individuals. An example of this is the Goliath frog (*Conraua goliath*), which tends to display high levels of aggression. Fights over resources put subordinate individuals at risk of immediate or prolonged death [21,26].

Furthermore, a lack of environmental stimuli, such as visual, auditory, gustatory, and olfactory cues, is suggested to have an impact on the development and quality of life of frogs and toads [25,26]. The direct result of this impact on individual well-being can trigger a range of pathologies with various manifestations, including anorexia, lethargy, aggression, low reproductive success, and mortality, among others [12,22].

## 5. Conclusions

Currently, behavioral research in the order Anura is limited due to the vast diversity of this species, which hinders the generalization of findings across the taxon. Additionally, many people perceive amphibian behavior as innate and driven by physiological factors and activities like foraging, courtship, and environmental cues such as temperature, light, humidity, and substrate type. Consequently, studies have primarily focused on individual shelter conditions.

Most research efforts have concentrated on the species of the *Xenopus* genus, benefiting from its extensive history as a laboratory animal and its contribution to experimental designs. However, investigations on enrichment have also been conducted in the frog genus *Dendrobates*. These studies have revealed that individuals can actively select certain accessories or additions, indicating their capacity to choose spaces that enhance their comfort and, in some cases, influence their development.

Nevertheless, establishing welfare indicators that capture the potential need for more complex forms of enrichment involving emotional states remains elusive. Currently, employed indicators only primarily detect negative or compromised welfare states that manifest when individuals experience developmental issues or illness.

It is important to highlight that certain behaviors have been described as potentially associated with cognitive stimuli-induced excitement, such as conspecific vocalizations or visual stimuli-like videos. However, further research is warranted to explore the practical implementation of such enrichment strategies in captive individuals, particularly those kept as pets. Lastly, recent discoveries have suggested that enrichment may involve learning and conditioning in amphibians, although the extent and nature of their learning capabilities remain uncertain. Therefore, substantial efforts are still required to advance our understanding of their environmental conditions and basic cognitive processes to propel future progress.

**Author Contributions:** Conceptualization, E.C.P. and R.I.M.R.; methodology, E.C.P. and R.I.M.R.; investigation, E.C.P.; data curation, E.C.P.; writing-original draft preparation, E.C.P.; writing-review and editing, R.I.M.R.; supervision, R.I.M.R. All authors have read and agreed to the published version of the manuscript.

**Funding:** This research received no external funding.

**Institutional Review Board Statement:** Not applicable.

**Data Availability Statement:** No new data were created or analyzed in thus study.

**Conflicts of Interest:** The authors declare no conflict of interest.

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
