# Peer review of "Welfare Indices in Anurans under Human Care"

_2673-5636, doi:10.3390/jzbg4030043_

Round 1

Reviewer 1 Report

I liked the subject of the research. However, I think the presentation/organization and the flow of information may be improved and that, I think, would make the paper more sound. My major concern is with the presentation of the results and the text of the discussion. I think that a great deal of the discussion would better serve as a presentation of the papers found by their search. In this regard a list of the species considered, their geographic distribution, their use in research and how these papers are relevant to the subject of "husbandry" (in general) and welfare of the anurans, would be better explored and more clearly presented. So, I suggest that the research is sound, but the paper would be better presented if a major reorganization be performed.

Author Response

Reviewer 1

  1. Union of the first and second paragraph of the introduction
  2. The table of references was replaced by a table that includes the results obtained from the methodology with relevant information about them.
  3. Changed: “This perception has led to a bias in the development of animal welfare plans for these species [10]. As a result, the management of captive amphibians primarily focuses 102 on meeting their basic survival requirements such as hydration, temperature, and food 103 [11, 12, 10].” Final result “This perception has led to a bias in the development of animal welfare plans for species in this group [10]. As a result, the management of captive amphibians primarily focuses on meeting their basic survival requirements such as hydration, temperature, and food [10 - 12].”

Reviewer 2 Report

Here are my major concerns with this MS. Since this is a review, I think the volume is not sufficient. The Introduction, Material and methods, Results and Discussion can be expanded. I think it would be nice if the author sum up all the commented in the Discussion section welfare indices into a table in the Results section. Currently it is very vague. The references need a complete re-formating. See my comments in the text.

Some spell-checking would be nice. Overall the English language is OK.

Author Response

  1. Word “order” and key words: not captalized
  2. The aim was added to the last paragraph of the introduction.
  3. Changed: The search encompassed the following reputable electronic databases.
  4. The selection criteria for the articles included in this review is more detailed.
  5. “Not necessary. People already know, what ethology and ethograms are.” We decided to keep this paragraph in the text, because we think that is not correct to assume that everyone reading this paper know what it is.
  6. Done: Cikanek et. al. Name of the authors followed by the number from the references list in square brackets is a given.
  7. Changed the reference to Vancouver.

Reviewer 3 Report

Please check the attached file

Author Response

  1. Added: “principally”.
  2. Add “salamenders”. We decided not to add this word because the cited text and the context are only about Mexican axolotls.
  3. Replaced: “certain species of the anura order” by “certain species of amphibians, and particularly of anurans”, because previosuly you have also mentioned examples of caudata species. Atendido
  4. Done: In anurans.
  5. Table 1: changed “comportamiento” by “behavoiur”.
  6. Added a brief introductory paragraph for the discusión section.
  7. Done: Connect this paragraph with what it is know in anurans. Atendido
  8. Done: Closing paragraph is missing for this section.
  9. Done: Dendodrates azureus and dendrobates tictoruis are synonyms.
  10. Done: I think that the current name is pelophyax ridibundus.
  11. Done: Like in other animal clades, anuran amphibians kept in captivity.

Round 2

Reviewer 2 Report

The authors agreed to all remarks made by me.

Author Response

Modifications done round two:

  1. Keywords changed to lower case.
  2. Changed the proper name of the institution.
  3. The table was modified according to the comments, some of the columns were eliminated and the information about reference was put together.